# The Dental Aesthetic Index and Its Association with Dental Caries, Dental Plaque and Socio-Demographic Variables in Schoolchildren Aged 12 and 15 Years

**DOI:** 10.3390/ijerph18189741

**Published:** 2021-09-16

**Authors:** Paula Fernández-Riveiro, Nerea Obregón-Rodríguez, María Piñeiro-Lamas, Almudena Rodríguez-Fernández, Ernesto Smyth-Chamosa, María Mercedes Suárez-Cunqueiro

**Affiliations:** 1Department of Psychiatry, Radiology, Public Health, Nursing and Medicine, Medicine and Dentistry School, Universidade de Santiago de Compostela, 15782 Santiago de Compostela, Spain; paula.fernandez.riveiro@usc.es (P.F.-R.); ernesto.smyth@usc.es (E.S.-C.); 2Department of Surgery and Medical Surgical Specialties, Medicine and Dentistry School, Universidade de Santiago de Compostela, 15782 Santiago de Compostela, Spain; nereaobr@gmail.com; 3Consortium for Biomedical Research in Epidemiology and Public Health (CIBER of Epidemiology and Public Health, CIBERESP), Instituto de Salud Carlos III, 28029 Madrid, Spain; maria.pineiro@usc.es; 4Health Research Institute of Santiago de Compostela (IDIS), University Clinical Hospital of Santiago de Compostela (CHUS/SERGAS), 15782 Santiago de Compostela, Spain

**Keywords:** dental aesthetic index, schoolchildren, dental caries, dental plaque, prevention, oral health, paediatric dentistry

## Abstract

The Dental Aesthetic Index (DAI) was determined in 12- and 15-year-old schoolchildren to ascertain the prevalence of malocclusion and to assess its association with dental caries experience, dental plaque accumulation, and socio-demographic variables. We performed a cross-sectional study with a stratified two-stage sampling design. An oral health survey and oral examination were conducted, and socio-demographic data were recorded. The sample comprised 1453 schoolchildren aged 12 (868) and 15 (585). These two samples were analyzed separately because statistically significant differences were found: the 12-year-old age group displayed a higher frequency of schoolchildren who attended state-run public schools (*p* = 0.004) and belonged to a lower social class (*p* = 0.001); the 15-year-old age group registered higher levels of caries (*p* = 0.001) and lower levels of dental plaque (*p* < 0.001). The malocclusion was 9.5% higher (*p* = 0.001), and the global mean DAI score was likewise higher among the 12-year-olds (*p* < 0.001). The multivariate regression analysis not only showed that caries and dental plaque were the variables that were the most strongly associated with malocclusion, but that caries (OR = 1.5) and dental plaque (OR > 2) were also risk factors for malocclusion in both groups. In conclusion, this study revealed a higher prevalence of malocclusion and dental plaque at age 12. A higher risk of caries and dental plaque was found to be related to the presence of malocclusion in both age groups.

## 1. Introduction

Malocclusion is generally ranked as the third highest oral health priority worldwide, due to its high prevalence and functional and psychosocial consequences [1,2]. Some consequences of malocclusion are bite and phonetic problems, temporo-mandibular dysfunction, and altered dental appearance affecting psychosocial well-being, self-esteem, and social interactions [2,3,4]. Although the degree of malocclusion severity can be ascertained by different indices [5,6], since its introduction in 1986, the Dental Aesthetic Index (DAI) has been widely used in various epidemiological studies to estimate the prevalence of malocclusion and orthodontic treatment needs [7,8]. Indeed, after only one year of being issued, the World Health Organization (WHO) included it as a recommended method for assessing malocclusion [9].

The DAI is based on a mathematical equation, which yields a score by summing the values of occlusal measurements associated with malocclusion (spacing, molar class, open-bite, crowding, irregularity, overjet, missing teeth) [10]. The DAI is reliable, objective, easy to use, and provides a large quantity of information on the malocclusion type and its severity [3,11]. Although the DAI equation has certain limitations, such as the underdiagnosis of deep and crossbite malocclusions, it is convenient due to its clinical and research applications [7,10]. The DAI is not only suitable for evaluating the severity of malocclusion and its consequences on oral health in the community, but it is also important for assessing treatment priorities [12].

Some types of malocclusion may be related to difficult dental plaque removal, which can facilitate the occurrence of gingival and dental pathology [13]. In such a case, early orthodontic treatment may be indicated to avoid oral pathology and to reduce the negative influence on growth and occlusal disorders [13].

Caries is one of the most prevalent diseases worldwide [14,15,16] and is the most frequent reason for receiving dental treatment among children [17]. Its aetiology is multifactorial and biofilm-mediated, but the dental plaque accumulation of fermentable carbohydrates is considered a key causal factor [18]. Oral hygiene is considered an essential factor for oral health [19]. Moreover, a higher incidence of malocclusion has also been related to missing teeth due to caries [20].

12- and 15-year-old adolescents represent a very important study group in epidemiological surveys of caries for several reasons, including the fact that this it is easy access to this population at schools; they are undergoing the final stage of permanent teeth eruption; there is the possibility of analyzing the first years of permanent dentition in the oral cavity; and this period is the beginning of self-made decisions about diet and hygiene [2,9].

The aims of this study were to ascertain the prevalence of malocclusion according to the DAI and to analyse its association with dental caries experience, dental plaque accumulation, and socio-demographic variables among 12- and 15-year-old schoolchildren.

## 2. Materials and Methods

### 2.1. Study Design and Sample Selection

An epidemiological oral health survey of random samples of schoolchildren aged 12 and 15 from north-west Spain (Galician Regional Authority) was conducted in accordance with the international guidelines established for this type of survey by the WHO [9]. We conducted a descriptive, cross-sectional study. The target size of the study population was estimated to be 23,500 pupils aged 12 and 20,000 pupils aged 15 attending a total of 485 schools that impart compulsory secondary education in this region. This information was obtained from the list of registered schools and pupils supplied by the Regional Authorities.

To select the sample, we used data sourced from previous Galician oral health surveys. We calculated a random sample of all secondary schools (both state-run public and private), which were stratified by province and size of town (≥20,000 or <20,000 inhabitants). In the sample, one classroom was randomly selected at each school; since the sample included 60 schools, this yielded 60 classrooms of pupils who were 12 years of age and 60 classrooms of pupils who were 15 years of age. In the case of the group that was 12 years of age, all of the individuals in each classroom were included; in the case of the group that was 15 years of age, a total of 12 individuals were randomly selected in each classroom. For schoolchildren to be eligible, the inclusion criteria were the following: (1) be aged 12 or 15 years old, (2) be present at school on the day of the examination; and (3) have written informed consent signed by their parents. The sample size, which was 1055 individuals who were 12 years of age and 788 individuals who were 15 years of age, was calculated with a 95% confidence level and an absolute error of 3.5% [21]. The calculated sample size included an additional 10% of individuals to offset any expected missing pupils and to compensate for a design effect, which we assumed to be 1.5.

To ensure greater representativeness, the sample was weighted. To achieve this, we first defined the design weights, taking the sample selection method into account. Each subject was then reweighted to adjust the distribution of the sample to the population of Galicia who were of that age, gender, and province.

The exclusion criteria that were applied were as follows: schools with fewer than ten pupils; pupils with gross facial asymmetry; pupils with developmental deformities; and pupils receiving orthodontic treatment at the time of the examination.

### 2.2. Data-Collection and Calibration

Data were collected by five purpose-trained work teams consisting of one dentist and one dental hygienist. Oral examination and socio-demographic data were collected from each pupil at the school. The oral examination was performed by the dentist in a seated position using basic oral examination tools [22,23]; the dental hygienist filled out the clinical examination form at the same time.

Initial calibration training was given in order to standardize the research methodology. To ensure the validity and reliability of results, inter- and intra-examiner calibrations were conducted to assess the diagnostic agreement between the examination teams and an external “benchmark” calibrator [24].

The DAI was used to record a number of parameters of occlusal features relating to tooth position as well as to the relationship between the maxillary and mandibular arches. The final DAI score classifies malocclusion into four categories, each of which are linked to an orthodontic treatment need: a score of ≤25 indicates normal or minor malocclusion (no treatment needed); a score of 26 to 30 represents moderate/definite malocclusion (elective treatment); a score of 31 to 35 indicates severe malocclusion; and a score of ≥36 indicates a very severe (handicapping) malocclusion. Furthermore, any DAI score of ≥31 is considered to require treatment [10]. Dental malocclusion was classified as null, moderate, severe, or very severe according to the DAI score [7]. In addition, the number of decayed, missing, and filled permanent teeth (DMFT) was noted in accordance with the WHO guidelines [9]. Caries were recorded as a cavitated lesion, and the presence of caries was considered as affected (DMFT > 0) or not affected (DMFT = 0). Oral hygiene was assessed by the variable dental plaque accumulation, with the absence/presence of dental plaque being evaluated visually by a periodontal WHO probe on the buccal surface of six teeth: first molars in both arches (16, 26, 36, 46) and upper and lower central incisors of one side (21, 41). The following four categories were listed: absence of dental plaque; plaque in the gingival border; plaque in 1/3 of the gingival border; and plaque in more than 1/3 of the gingival border. These variables and their categories are shown in Table 1.

The socio-demographic variables analyzed were age, gender, type of school (state-run public or private), and residential setting (urban ≥20,000 inhabitants; rural <20,000 inhabitants).

### 2.3. Data Analysis

Frequency distribution was used to describe the characteristics of the sample. Differences between age groups were evaluated using Pearson’s Chi-square test. The mean DAI scores and the prevalence of malocclusion as per the DAI score were calculated for each age group, with the Student’s t-test being used for the comparison of the means between the continuous variables. In order to examine the association between malocclusion on the one hand and socio-demographic (gender, type of school, residence) and oral health variables (caries prevalence, oral hygiene) on the other, we performed a logistic regression analysis, taking DAI scores higher than 26 as indicative of malocclusion for this purpose. To construct the models, we first performed a bivariate analysis with the exposure variables and potential confounding variables and then fitted a multivariate logistic regression model that included all of those independent variables that had been proven to have statistical significance lower than 0.2 in the bivariate analysis. To obtain the best mathematical model, independent variables with a higher level of statistical significance were eliminated from this model, provided that the coefficients of the main exposure variables had not changed by more than 10% and that the Akaike Criterion had improved. All data were analyzed independently for each age group. To analyse the calibration agreement, the intraclass correlation coefficient index was calculated for the DAI scores, and the Kappa coefficient was used for the malocclusion grades defined according to both indices. All *p*-values were two-sided, with *p*-values of 0.05 or less being deemed statistically significant. All statistical analyses were performed using the R Survey Package (version 4.0.3, The R Foundation for Statistical Computing, c/o Institute for Statistics and Mathematics, Vienna, Austria).

### 2.4. Ethical Considerations

As this study formed part of an evaluation of the oral public health services provided by the Galician Regional Authority (*Xunta de Galicia),* approval by the Galician Clinical Research Ethics Committee was not required. Participation was voluntary. Once the schools agreed to participate, an information sheet and informed written consent form was sent to each family. Signed written consent by parents/legal tutor was required in order to participate in the study. All data were analyzed anonymously to ensure confidentiality.

## 3. Results

The final sample included 1453 schoolchildren, comprising 868 pupils (418 males and 450 females) who were 12 years of age and 585 pupils (271 males and 314 females) who were 15 years of age. Application of the exclusion criteria resulted in ten and five losses from the groups of 12-year-olds and 15-year-olds, respectively, due to absence on the day of examination, and 176 and 198 losses in the 12- and 15-year-old groups, respectively, due to the presence of orthodontic appliances at the time of examination.

The distribution of socio-demographic and oral health data for the two age groups is shown in Table 1. The type of school displayed statistically significant differences (*p* < 0.001), i.e., 65.3% of pupils in the sample of pupils who were 12 years of age attended a state-run public school versus 57.2% in the sample of pupils who were 15 years of age. Statistical differences were also found between the two age groups in terms of social class, in that there were more students of a lower social class in the 12-year-old group than in the 15-year-old group. Information about differences between age groups are available at Appendix A

The percentage of individuals free of dental caries was 59.1% (95% CI, 55.5–62.7) at 12 years of age versus 49.4% (95% CI, 45.1–53.6) at 15 years of age, a statistically significant difference (*p* = 0.001). With regard to caries experience (DMFT > 0), the schoolchildren who were 15 years of age had values that were 10% higher. Dental plaque was observed in 80.4% (95% CI, 77.4–83.2) of the pupils who were 12 years of age and in 64.4% (95% CI, 60.3–68.5) of pupils who were 15 years of age (*p* < 0.001). Pupils who were 12 years of age showed dental plaque accumulation ≥1/3 gingival coverage more frequently than pupils who were 15 years of age did (*p* < 0.001).

A total of 36.7% (95% CI, 33.8–39.6) of the schoolchildren presented with malocclusion, as did 29.3% (95% CI, 25.5–33.4) of the pupils in the 15-year-old group and 38.8% (95% CI, 35.2–42.4) of the pupils in the 12-year-old group. The frequency of malocclusion was 9.5% higher in the 12-year-old group, displaying statistically significant differences (*p* = 0.001). The mean DAI scores were 24.78 (95% CI, 24.26–25.30) for the 12-year-olds and 22.43 (95% CI, 21.90–22.96) for the 15-year-olds, as seen in Table 2. The global mean DAI score was 2.35 (95% CI, 1.61–3.09) points higher in the group of 12-year-olds, showing statistically significant differences (*p* < 0.001).

In terms of the DAI category, 17.2% of the pupils (95%CI, 14.9–19.7) displayed severe/very severe malocclusion, and 19.5% (95% CI, 17.2–21.9) displayed moderate malocclusion; this was the case in both groups overall, with significant differences (*p* < 0.001) between the two age groups.

The distribution of socio-demographic and dental data for children with no malocclusion, moderate malocclusion, and severe/very severe malocclusion is shown in Table 3 and Table 4: first, 19.8% of 12-year-olds (95% CI, 17.0–22.9) had severe/very severe malocclusion versus 8.1% (95% CI, 5.9–10.8) of 15-year-olds; and second, the percentage of schoolchildren with no malocclusion was statistically higher in the 15-year-old age group, 70.7% (95% CI, 66.6–74.5), than in the 12-year-old age group, 61.2% (95% CI, 57.6–64.8).

Significant differences were found between the DAI categories and caries experience (DMFT > 0). Among the 12-year-old group, 26.4% (95% CI, 21.5–32) of the pupils in the group with caries had severe/very severe malocclusion versus only 15.3% (95% CI, 12.2–19) in the group without caries. Among the 15-year-old group, 65.6% (95% CI, 59.5–71.1) of the pupils in the group with caries had no malocclusion versus 75.9% (95% CI, 70.1–80.9) in the group without caries.

Statistically significant differences were likewise found between different levels of malocclusion and dental plaque accumulation in both age groups. In the 12-year-old group without dental plaque, 11.5% (95% CI, 7.1–18.1) had severe/very severe malocclusion, while 27.2% (95% CI, 21.8–33.4) of the pupils with dental plaque in gingival border or ≥1/3 gingival had severe/very severe malocclusion. In the 15-year-old group without dental plaque, 82.12% (95% CI, 75.8–87) had no malocclusion, whereas 66% (95% CI, 56.3–74.6) of pupils with dental plaque in gingival border or ≥1/3 gingival had no malocclusion.

Furthermore, there were significant differences regarding the type of school in the 12-year-old group.

Table 5 and Table 6 show the association between malocclusion (DAI > 25) and socio-demographic variables, caries experience, and dental plaque. The multivariate regression analysis showed that caries experience and dental plaque accumulation were the variables that were the most strongly associated with malocclusion (DAI > 25) in both age groups. In the 12-year-old group, caries experience posed a 58% risk (95% CI, 15–117) of having malocclusion, and dental plaque accumulation on the gingival border (OR = 1.99; 95% CI, 1.28–3.12) or on ≥1/3 gingival (OR = 2.66; 95% CI, 1.67–4.23) were also risk factors of malocclusion. Similarly, in the 15-year-old group, the presence of caries posed a 53% (95% CI, 3–127) risk of having malocclusion, and dental plaque accumulation on the gingival border (OR = 2.52; 95% CI, 1.58–4.03) or on ≥1/3 gingival (OR = 2.18; 95% CI, 1.24–3.84) were again found to be risk factors for malocclusion. Moreover, in the 12-year-old group, there was an association between malocclusion and the type of school, with pupils who attended state-run public schools having a higher risk of malocclusion (OR = 1.54; 95% CI, 1.11–2.15). There were no significant differences between the respective DAI scores in the two age groups in terms of gender, socio-economic status, or residential setting.

Table 7 shows the distribution frequency of the DAI components. The most frequent disorders among the 12-year-olds were an anterior maxillary overjet ≥4 mm and the largest anterior maxillary and mandibular irregularity of 1–2 mm. The most frequent DAI component disorders among 15-year-olds were the largest anterior maxillary and mandibular irregularity of 1–2 mm, crowding in one incisal segment, and an anterior maxillary overjet ≥4 mm.

The intraclass correlation coefficient and Kappa coefficient calculated for both interobserver and intraobserver errors indicated noticeably good overall agreement. Appendix A are available (Appendix A).

## 4. Discussion

Our study’s main finding was that caries experience and dental plaque are risk factors for malocclusion in both age groups, displaying the strongest association with dental plaque in the 12-year-old group. While dental plaque accumulation was higher among the 12-year-olds, caries experience was higher among the 15-year-olds. A higher risk of malocclusion was found to be associated with attendance at state-run public schools at 12 years of age. No associations were found between the DAI and other socio-demographic variables in either age group.

Some studies have reported on the multifactorial aetiology of caries [25,26]. A systematic review conducted in 2012 found that there were no high-quality studies to indicate the possible association between dental crowding and caries [27]. In contrast, other studies have reported a significant association between malocclusion and caries rates or dental plaque [28,29,30,31,32], as observed by our study. Singh et al. [31] also reported a positive correlation between severe/handicapping malocclusion and dental caries among 12-year-old Indian schoolchildren. Some authors have only observed a correlation between specific types of malocclusion, such as mandibular overjet and posterior cross-bite, and increased caries risk in mixed dentition [33]. Similarly, while our study observed an improvement in dental plaque accumulation at age 15, it also observed higher values of caries experience. The reason why malocclusion could increase the occurrence of caries might be an accumulation of biofilm over long periods of time in relatively inaccessible areas that are difficult to clean [34]. Some studies also suggest a relationship between malocclusion and gender, social class, residential setting, and type of school [8,11,35]. Our study observed an association in the 12-year-old age group between the type of school and malocclusion, but there was no association with any of the other variables. The incidence of minor or no malocclusion was found to be higher in private schools, possibly for financial reasons, due to the fact that orthodontic treatment is exclusively provided by the private sector in Spain. Government financial support for orthodontic treatment should therefore be urged for the most severe malocclusions.

We observed that the frequency of malocclusion did not vary between urban and rural residential settings, a finding that is in line with other studies [36]. This could be due to an increase in the number of dental offices in recent years as a result of urban sprawl towards rural areas. Nevertheless, studies undertaken in other developed countries report major differences between urban and rural areas, probably due to differences in access to treatment [8,37].

The mean DAI scores obtained by our study are similar to those found in other populations for both age groups [36,38], with these scores being higher in the 15-year-old group. This could be accounted for by the accumulation of risk factors over a longer period of time. Whereas the DAI scores that we recorded in the 12-year-old group were very similar to those reported by other studies [11,38,39], those recorded in the 15-year-old group were lower than those reported elsewhere [36] and were closer to the 18% found among the 12-year-old school-goers in India [31] or to the 22.6% found in Nigerian secondary schoolchildren aged 12 to 18 [40].

In this study, the mean DAI score was lower in the group of pupils who were 15 years of age. This reduction has been observed in other studies [35,41] and could be attributable to individuals who underwent orthodontic treatment before the age of 15. In contrast, Anita et al. observed a greater presence of malocclusion in older individuals [12]. The frequency of handicapping malocclusion found in our study was similar to that reported by most studies for both age groups [11,38,39,41].

With respect to the DAI components, crowding and maxillary overjet ≥3 mm are known to be the main occlusal features observed in different studies [33,41,42]. In our case, maxillary irregularity ≥3 mm and maxillary overjet ≥4 mm were observed to be the most frequent occlusal traits, with similar findings being described in the literature [30,41,42]. Mandibular overbite and frontal open bite had a similar prevalence to that reported in the literature [33]. The frequency of midline diastema, however, was found to be lower than in other studies [11,40].

The strengths of this study lie in its sample size and representativeness of the study population, the strong associations found (OR) in relation to malocclusion, the presence of dental plaque and caries experience, and the clinical implications that these findings have for oral health. Another strength of our study was the use of data obtained by trained clinical examiners, which represents greater objectivity than data obtained exclusively through self-completed questionnaires. Likewise, the collection of data over several years strengthens our results.

Our study is not free of limitations. First, the cross-sectional study design cannot determine whether caries is a cause or consequence of malocclusion because both variables were analyzed at the same time. Existing relationships with other caries risk factors described in the literature that we have not included in our present research, such as income, the use of fluorides, dietary habits, or anxiety about dental treatment, could also be included in future research. Similarly, data on the history of previous orthodontic treatment should be collected by future studies in order to better explain variations in the DAI.

## 5. Conclusions

Whereas the prevalence of malocclusion and dental plaque was higher among the 12-year-olds, experience of caries was higher among the 15-year-olds. However, no noteworthy associations were found in terms of socio-demographic variables. Caries and dental plaque were observed to be risk factors for malocclusion in both age groups. This finding has definite clinical implications for the prevention of dental caries and oral health promotion in young adolescents. Accordingly, future research should focus on designing a longitudinal study to confirm the cause–effect relationship of the variables that were studied.

## Figures and Tables

**Table 1 ijerph-18-09741-t001:** Distribution of socio-demographic and oral health variables in the samples of 12- and 15-year-old schoolchildren.

Variables	Age 12			Age 15			*p*-Value ^b^
	n	% ^a^	95% CI	n	% ^a^	95% CI	
Sex							
Male	418	47.8	44.1–51.5	271	46.5	42.3–50.8	0.6700
Female	450	52.2	48.5–55.9	314	53.5	49.2–57.7	
Social class							
High	111	13.2	10.8–15.8	84	14.8	11.9–18.0	0.009
Medium	350	40.2	36.6–43.9	276	47.4	43.1–51.7	
Low	404	46.6	43.0–50.3	224	37.9	33.8–42.1	
Residence							
Urban area	508	53.5	49.8–57.2	371	55.5	51.1–59.8	0.496
Rural area	360	46.5	42.8–50.2	214	44.5	40.2–48.9	
Type of school							
Public	565	65.3	61.7–68.8	375	57.2	52.8–61.4	0.004
Non-public	303	34.7	31.2–38.3	210	42.8	38.6–47.2	
Caries experience							
No	517	59.1	55.5–62.7	294	49.4	45.1–53.6	0.001
Yes	351	40.9	37.3–44.5	291	50.6	46.4–54.9	
Dental plaque							
Absence	179	19.6	16.8–22.6	207	35.6	31.5–39.8	<0.001
On gingival border	400	46.8	43.1–50.5	256	44.0	39.8–48.3	
On 1/3 gingival	211	24.4	21.3–27.7	100	16.7	13.7–20.1	
On >1/3 gingival	77	9.2	7.2–11.5	22	3.7	2.3–5.5	

^a^ The percentages were calculated taking the weighting of the sample into account, so they do not match the raw percentages. ^b^ Comparison between 12- and 15-year-olds using Ji^2^ test, adjusted by sample design.

**Table 2 ijerph-18-09741-t002:** Mean Dental Aesthetic Index (DAI) in 12- and 15-year-old schoolchildren.

Malocclusion Severity	Age 12		Age 15	
	Mean	95% CI	Mean	95% CI
Global ^a^	24.78	24.26–25.30	22.43	21.90–22.96
Non-malocclusion	20.30	19.99–20.60	19.36	19.03–19.69
Moderate malocclusion	27.66	27.43–27.89	27.62	27.35–27.88
Severe malocclusion	32.87	32.59–33.15	32.39	31.83–32.96
Very severe malocclusion	40.99	39.78–42.20	39.32	37.87–40.78

^a^ Difference between mean DAI at ages 12 and 15: 2.28; 95% CI, 1.18–3.38; *p* < 0.001.

**Table 3 ijerph-18-09741-t003:** Presence and severity of malocclusion by socio-demographic and oral health variables and estimation of the association between the presence of malocclusion and the different variables in 12-year-old schoolchildren.

Variables	Null		Moderate		Severe/Very Severe		*p*-Value
	%	95% CI	%	95% CI	%	95% CI	
Global	61.2	57.6–64.8	18.9	16.2–21.9	19.8	17.0–22.9	
Sex							
Male	58.5	53.2–63.7	19.6	15.7–24.1	21.9	17.8–26.8	0.31
Female	63.7	58.7–68.4	18.4	14.8–22.6	17.9	14.3–22.2	
Social class							
High	61.8	51.5–71.1	20.1	13.2–29.2	18.2	11.4–27.7	0.698
Medium	64.1	58.3–69.5	16.7	13.0–21.3	19.1	14.8–24.3	
Low	59.0	53.6–64.2	20.7	16.7–25.5	20.3	16.3–25.0	
Residence							
Urban area	63.3	58.5–67.9	16.8	13.5–20.7	19.9	16.2–24.2	0.278
Rural area	58.8	53.2–64.2	21.4	17.2–26.3	19.8	15.6–24.7	
Type of school							
Public	57.2	52.7–61.7	22.3	18.8–26.3	20.5	17.0–24.4	0.003
Non-public	68.7	62.6–74.2	12.6	9.1–17.1	18.7	14.2–24.2	
Caries experience							
No	66.4	61.8–70.7	18.3	15.0–22.2	15.3	12.2–19.0	<0.001
Yes	53.7	47.9–59.5	19.9	15.7–24.8	26.4	21.5–32.0	
Dental plaque							
Absence	77.0	69.5–83.1	11.5	7.5–17.4	11.5	7.1–18.1	<0.001
On gingival border	60.8	55.4–66.0	21.3	17.3–26.1	17.8	14.0–22.4	
On ≥1/3 gingival	52.8	46.4–59.2	20.0	15.4–25.5	27.2	21.8–33.4	

**Table 4 ijerph-18-09741-t004:** Presence and severity of malocclusion by socio-demographic and oral health variables and estimation of the association between the presence of malocclusion and the different variables in 15-year-old schoolchildren.

Variables	Null		Moderate		Severe/Very Severe		*p*-Value
	%	95% CI	%	95% CI	%	95% CI	
Global	70.7	66.6–74.5	21.2	17.8–24.9	8.1	5.9–10.8	
Sex							
Male	68.4	62.1–74.1	21.1	16.4–26.7	10.5	7.0–15.5	0.177
Female	72.6	67.0–77.6	21.4	16.9–26.7	6.0	3.8–9.5	
Social class							
High	76.6	65.3–85.0	17.7	10.5–28.4	5.7	2.1–14.6	0.401
Medium	72.6	66.5–77.9	19.2	14.7–24.6	8.3	5.2–12.8	
Low	65.9	58.9–72.2	25.3	19.6–32.0	8.9	5.6–13.7	
Residence							
Urban area	72.3	67.2–76.8	20.7	16.7–25.4	7.0	4.7–10.3	0.554
Rural area	68.7	61.7–74.9	21.9	16.6–28.4	9.4	5.9–14.8	
Type of school							
Public	71.6	66.5–76.2	21.1	17.1–25.9	7.3	5.0–10.5	0.742
Non-public	69.4	62.4–75.7	21.4	16.1–27.9	9.2	5.6–14.6	
Caries experience							
No	75.9	70.1–80.9	17.1	12.9–22.3	7.0	4.3–11.2	0.041
Yes	65.6	59.5–71.1	25.3	20.3–30.9	9.2	6.2–13.4	
Dental palque							
Absence	82.1	75.8–87.0	13.7	9.3–19.6	4.2	2.2–8.0	0.001
On gingival border	63.6	57.1–69.6	26.7	21.3–32.9	9.7	6.4–14.6	
On ≥1/3 gingival	66.0	56.3–74.6	22.7	15.7–31.7	11.2	6.1–19.8	

**Table 5 ijerph-18-09741-t005:** Malocclusion risk in 12-year-old schoolchildren using bivariate and multivariate logistic regression models.

Variables	Malocclusion Yes/No					
	Bivariate Model			Adjusted Model		
	OR	95% CI	*p*-Value	OR	95% CI	*p*-Value
Sex						
Male	1.24	0.92–1.68	0.157			
Female	1					
Social class						
High	1					
Medium	0.9	0.56–1.46	0.682			
Low	1.12	0.7–1.8	0.628			
Residence						
Urban area	1					
Rural area	1.21	0.89–1.64	0.221			
Type of school						
Public	1.64	1.18–2.27	0.003	1.54	1.11–2.15	0.011
Non-public	1			1		
Caries experience						
No	1			1		
Yes	1.7	1.25–2.32	0.001	1.58	1.15–2.17	0.004
Dental plaque						
Absence	1			1		
On gingival border	2.15	1.38–3.35	0.001	1.99	1.28–3.12	0.003
On ≥1/3 gingival	2.99	1.88–4.74	<0.001	2.66	1.67–4.23	<0.001

**Table 6 ijerph-18-09741-t006:** Malocclusion risk in 15-year-old schoolchildren using bivariate and multivariate logistic regression models.

Variables	Malocclusion: Severe or Very Severe/No or Moderate					
	Bivariate Model			Adjusted Model		
	OR	95% CI	*p*-Value	OR	95% CI	*p*-Value
Sex						
Male	1.23	0.84–1.8	0.297			
Female	1					
Social class						
High	1					
Medium	1.24	0.67–2.28	0.499			
Low	1.69	0.91–3.15	0.096			
Residence						
Urban area	1					
Rural area	1.19	0.81–1.75	0.381			
Type of school						
Public	0.9	0.61–1.33	0.602			
Non-public	1					
Caries experience						
No	1			1		
Yes	1.65	1.12–2.44	0.011	1.53	1.03–2.27	0.037
Dental plaque						
Absence	1			1		
On gingival border	2.62	1.65–4.17	<0.001	2.52	1.58–4.03	<0.001
On ≥1/3 gingival	2.35	1.35–4.1	0.003	2.18	1.24–3.84	0.007

**Table 7 ijerph-18-09741-t007:** Frequency distribution of malocclusion traits according to DAI components in 12- and 15-year-old schoolchildren.

DAI Components	Age 12		Age 15		*p*-Value
	%	95% CI	%	95% CI	
Number of missing upper teeth (≥1)	6.8	5.1–8.9	1.8	0.9–3.3	<0.001
Number of missing lower teeth (≥1)	2.1	1.2–3.5	0.2	0–0.8	0.001
Crowding (incisal segments)					
No segment crowed	68.4	64.9–71.7	63	58.7–67.1	0.124
One segment crowed	22.5	19.6–25.7	27.2	23.4–31.2	
Two segments crowed	9.1	7.1–11.3	9.8	7.4–12.6	
Spacing in the incisal segments					
No spacing	72.9	69.6–76.1	83.8	80.4–86.8	<0.001
One segment spaced	24.5	21.4–27.7	14.1	11.2–17.3	
Two segments spaced	2.6	1.7–3.8	2.1	1.2–3.5	
Midline diastema, in mm (≥1)	18.3	15.6–21.3	10.2	7.8–13	<0.001
Largest anterior maxillary irregularity, in mm					
0	54.9	51.2–58.5	60.7	56.4–64.8	0.088
1–2	30.6	27.3–34	28.1	24.4–32.1	
≥3	14.5	12.1–17.2	11.2	8.6–14.2	
Largest anterior mandibular irregularity, in mm					
0	61.7	58.1–65.2	60.9	56.7–65	0.813
1–2	31.6	28.3–35.1	31.5	27.6–35.6	
≥3	6.6	4.9–8.7	7.6	5.5–10.1	
Anterior maxillary overjet, in mm					
0–3	63.2	59.6–66.7	75.4	71.5–79	<0.001
≥4	36.8	33.3–40.4	24.6	21–28.5	
Anterior mandibular overjet in mm (>0)	2.2	1.3–3.4	2.2	1.1–3.7	0.960
Vertical anterior open bite, in mm (>0)	2.7	1.7–4.1	3	1.8–4.6	0.758
Anteroposterior molar relationship, largest deviation from normal either left or right					
Normal	60.2	56.6–63.8	68.8	64.8–72.7	0.009
1/2 cusp either mesial or distal	35.6	32.1–39.2	27	23.4–30.9	
One full cusp or more either or mesial	4.2	2.9–5.9	4.1	2.5–6.4	

## Data Availability

The datasets used and analysed during the current study are available from the corresponding author upon reasonable request.

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
