# Peer review of "The Dental Aesthetic Index and Its Association with Dental Caries, Dental Plaque and Socio-Demographic Variables in Schoolchildren Aged 12 and 15 Years"

_ijerph, 2021, doi:10.3390/ijerph18189741_

Round 1
Reviewer 1 Report
The manuscript entitled “The Dental Aesthetic Index and its association with dental caries, dental plaque and socio-demographic variables in school-children aged 12 and 15 years” presents the association between Dental Aesthetic Index and dental caries, dental plaque, and socio-demographic variables in schoolchildren aged 12 and 15 years. The authors presented the significance of a strong association between the Dental Aesthetic Index and dental plaque in 12-year-old schoolchildren. Additionally, they have provided evidence that caries and dental plaque are considered risk factors for malocclusion when examining schoolchildren aged 12 and 15 years.
Major issues are listed in comments on Materials and Methods, and Discussion, whereas minor issues are provided in all other comments. Please, see below.
Major issues:
Comments on Materials and Methods:
Please provide explanations for:
- “We conducted a systematic, descriptive, cross-sectional study.”;
- Why have you considered all individuals aged 12 years and a random selection of 12 individuals aged 15 years?
- Could you please provide the sample size?
- Please use a synonym for “achieve” as it was twice used in line 93
- criteria of study subjects’ inclusion are not described. Please add the missing information.
- Galician oral health survey: in the study design and sample selection, you mentioned that data were obtained from the Galician oral health survey, which you only later describe in the study procedure and calibration subsection. However, this subsection is supposed to be about your study procedure and not about the study procedure of another study. Please revise the entire section accordingly.
- Subsection data-collection: As mentioned in the comments on the introduction, I suggest adding the explanation about DAI here.
- Subsection data-analysis: Please omit the first sentence as it was mentioned earlier in the article.
- Please explain why you have eliminated independent variables with a higher level of statistical significance.
- Please add a dot at the end of line 162.
Comments on Discussion:
- The discussion fits well with the aims of the study stated in the introduction.
- Comparison with the literature and the limitations of the study are suitable.
- Please revise lines 296-297, 309-312, 322-325.
- Please also provide a few statements about the strengths of the study.
- In line 322 you mention that “the significant associations observed cannot be fully explained.” Could you provide explanations for it to make it clearer to the reader why is it so?
- Please add information on why do you consider a limitation of the study the point that you have not analyzed “Existing relationships with other caries risk factors described in the literature.”
Minor issues:
Comments on Conclusion:
- The authors’ conclusions are suitable.
Comments on Title, Abstract, and Keywords:
- The title is suitable and accurately explains what the study was about.
- The abstract summarizes well the manuscript except for the conclusion that is missing. Please consider adding a conclusion statement, a sentence that indicates your contribution to the science, and what would be the next steps in the oral health area.
- The information included in the abstract is precise and follows the data of the manuscript.
- Keywords are suitable. Nevertheless, please capitalize the words that require it.
Comments on Introduction:
- The paragraph (lines 47-54) would better fit into the methods section as it is a description of the methods. Please consider moving the paragraph.
- Line 60: Please rewrite the sentence to read: ..., but IT is also important when it comes to assessing treatment priorities.
- Please rephrase the sentence in lines 69-70 to be more comprehensible.
- The aim in the abstract and introduction sections is not congruent. I suggest using the more detailed one in both sections.
Comments on Results and Figures:
- Could you please rephrase the sentence in lines 164-169 to be clearer?
- Data was presented and reported very nicely and has no further comments.
Comments on Acknowledgements:
- The acknowledgments section should be revised. Consider also writing in the past tense.
Comments on References:
- The authors of the manuscript cited relevant references that represent current knowledge in the oral health area. They have also cited a variety of research groups and, therefore, references could be helpful for other researchers. However, references should be cited before full stops or commas. Please revise all the references according to the Journal recommendations.
Author Response
Dear Editors and Reviewers,
Thank you very much for your letter and for the reviewers’ comments concerning our manuscript entitled “The Dental Aesthetic Index and its association with dental caries, dental plaque and socio-demographic variables in school-children aged 12 and 15 years”. Those comments are all valuable and very helpful for revising and improving our paper. Based on your comments and requests, we have made careful modifications on the original manuscript.
Reviewer 1
Major issues
Comments on Materials and Methods:
- “Please provide explanations for: “we conducted a systematic, descriptive, cross-sectional study”.”
Answer. We used the word systematic because we carried out several oral health surveys in Galicia (1995, 2000, 2005, 2010) following the same methodology but we have not reviewed the literature systematically. This is the reason why, following your indications, we have carefully reviewed this sentence, and now, we have changed our old sentence “We conducted a systematic, descriptive, cross-sectional study” by the new sentence: “We conducted a descriptive, cross-sectional study”. This was made in Materials and Methods section, line 82.
- “Why have you considered all individuals aged 12 years and a random selection of 12 individuals aged 15 years?”
Answer. We appreciate your interesting question. We considered a random selection of individuals aged 15 because of financial reasons, the main original population sample included in the study were the individuals aged 12. The sample of students aged 15 were included subsequently.
- “Could you please provide the sample size?”
Answer. You can find this information at Materials and Methods section, line 96-98: “The sample size, 1055 individuals aged 12 years and 788 aged 15 years, was calculated with a 95% confidence level and an absolute error of 3.5%”. After exclusion criteria “The final sample included 1453 schoolchildren, comprising 868 pupils (418 males and 450 females) aged 12 years and 585 pupils (271 males and 314 females) aged 15 years. Application of the exclusion criteria resulted in 10 and 5 losses in the groups aged 12 and 15 years, respectively, due to absence on the day of examination; and 176 and 198 losses in the 12- and 15-year-old groups respectively, due to the presence of orthodontic appliances at the time of examination.” Result section, lines 169-174.
- “Please use a synonym for “achieve” as it was twice used in line 93.”
Answer. As suggested, the first use of “achieve” has been dropped. In Materials and Methods section, line 101, the former wording, “To achieve greater representativeness, the sample was weighted” has thus been amended to read, “To ensure greater representativeness, the sample was weighted.”
- “Criteria of study subjects’ inclusion are not described. Please add the missing information.”
Answer. We really appreciate your suggestion, and, now, we have added the information about inclusion criteria in Material and Methods section, lines 92-94: “For schoolchildren to be eligible, the inclusion criteria were the following: 1) to be aged 12 or 15 years old, 2) to be present at school on the day of examination; and 3) to have written informed consent signed by their parents”
- “Galician oral health survey: in the study design and sample selection, you mentioned that data were obtained from the Galician oral health survey, which you only later describe in the study procedure and calibration subsection. However, this subsection is supposed to be about your study procedure and not about the study procedure of another study. Please revise the entire section accordingly.”
Answer. We appreciate your suggestion and we have changed the entire section, replacing the information about the study procedure accordingly. For that, we have modified the subsection study design and sample selection, and the subsection data collection and calibration. Therefore, the subsection study procedure and calibration has been removed, and the information has been replaced accordingly. You can see these changes in the Material and Methods section, lines 78-82, and 109-118: “An epidemiological oral health survey of random samples of schoolchildren aged 12 and 15 years from north-west Spain (Galician Regional Authority) was conducted in accordance with the international guidelines established for this type of survey by the WHO [9]. We conducted a descriptive, cross-sectional study.”
“2.2. Data-collection and calibration. Data were collected by five purpose-trained work teams consisting of one dentist and one dental hygienist. Oral examination and socio-demographic data were collected from each pupil at the school. The oral examination was performed by the dentist in a seated position using basic oral examination tools [22,23], with the subject’s respective clinical examination form being simultaneously completed by the dental hygienist. Initial calibration training was given in order to standardize the research methodology. To ensure the validity and reliability of results, inter- and intra-examiner calibrations were conducted to assess diagnostic agreement between the examination teams and an external “benchmark” calibrator [24].”
- “Subsection data-collection: As mentioned in the comments on the introduction, I suggest adding the explanation about DAI here.”
Answer. We have carefully reviewed the explanation about DAI, and, as suggested, we have removed the DAI explanation from the introduction, and we have added it on subsection data-collection. A brief explanation about DAI was kept in the introduction, in order to correctly introduce and explain the aim of the study. You can see these changes in Materials and Methods section, lines 120-126: “The DAI was used to record a number of parameters of occlusal features relating to tooth position, as well as the relationship between maxillary and mandibular arches. The final DAI score classifies malocclusion into 4 categories, each linked to an orthodontic treatment need: a score of ≤25 indicates normal or minor malocclusion (no treatment needed); a score of 26 to 30 represents moderate/definite malocclusion (elective treatment); a score of 31 to 35 indicates severe malocclusion, and a score of ≥36 indicates a very severe (handicapping) malocclusion. Furthermore, any DAI score of ≥31 is considered to require treatment [10].”
- “Subsection data-analysis: Please omit the first sentence as it was mentioned earlier in the article.”
Authors. We have followed your suggestion, and now, we have deleted that sentence. You can see these changes in Materials and Methods section, Data-analysis subsection.
- “Please explain why you have eliminated independent variables with a higher level of statistical significance.”
Answer. According to the statistical expert, the mathematical model becomes the best model to analyze the variables when the independent variables with a higher level of statistical significance are eliminated. Variables that influence ≤10% in the dependent variable are eliminated from the mathematical model. The explanation can be seen in Materials and Methods section, Data-analysis subsection, page 3, lines 152-154: “To obtain the best mathematical model, independent variables with a higher level of statistical significance were eliminated from this model, provided that the coefficients of the main exposure variables had not changed by more than 10% and that the Akaike Criterion had improved.”
- “Please add a dot at the end of line 162.”
Answer. We have carefully reviewed this sentence, and now, we have added a dot at the end of that current line.
Comments on Discussion
- “The discussion fits well with the aims of the study stated in the introduction.”
Answer. Thank you very much for this comment.
- “Comparison with the literature and the limitations of the study are suitable.”
Answer. We appreciate this comment.
- “Please revise lines 296-297, 309-312, 322-325.”
Answer. The indicated lines were revised, and some changes were made. You can see these modifications highlighted in yellow in the text, in Discussion section, page 10, lines 300-301, 313-316, and 330-332. We have changed the previous 296-297 (now, lines 302-303) “This could be due to the rural population has increased dental offices in recent years, as a result of urban sprawl towards rural areas” by the new sentence: “This could be due to an increase in the number of dental offices in recent years as a result of urban sprawl towards rural areas.” (lines 302-304). We have modified the old 309-312 “This reduction has been observed in other studies,[35,42] and could be accounted for by schoolchildren aged 12 to 15 years undergoing orthodontic treatment. In contrast, Anita et al.[12] observed an increase in malocclusion with age” by the new: “This reduction has been observed in other studies [35,42], and could be attributable to individuals who underwent orthodontic treatment before the age of 15. In contrast, Anita et al. observed a greater presence of malocclusion in older individuals [12] ” (lines 315-318). And, we have changed the previous 322-325: “Firstly, the significant associations observed cannot be fully explained, in that the current analysis is based on a cross-sectional design, thereby making it impossible to ascertain the order of occurrence of malocclusion and caries.” by the new text: “Firstly, the cross-sectional study design cannot determine whether caries is the cause or consequence of malocclusion.” (lines 332-333)
- “Please also provide a few statements about the strengths of the study.
Answer. We have added a few statements about the strengths of our study (lines 328-331): “The strengths of this study lie in its sample size and representativeness of the study population, the strong associations found (OR) in relation to malocclusion, presence of dental plaque and caries experience, and the clinical implications that these findings have for oral health.”
- “In line 322 you mention that “the significant associations observed cannot be fully explained.” Could you provide explanations for it to make it clearer to the reader why is it so?”
Answer. We appreciate your comment, and, now, we have modified the old sentence “the significant associations observed cannot be fully explained, in that the current analysis is based on a cross-sectional design, thereby making it impossible to ascertain the order of occurrence of malocclusion and caries”, (lines 332-333) by the new sentence: “Firstly, the cross-sectional study design cannot determine whether the caries is the cause or the consequence of malocclusion.”
- “Please, add information on why do you consider a limitation of the study the point that you have not analyzed “Existing relationships with other caries risk factors described in the literature”.”
Answer. Following your suggestions, we have modified the sentence in order to explain it better. As you can see, we have modified our old text “Existing relationships with other caries risk factors described in the literature (such as income, use of fluorides, dietary habits, anxiety about dental treatment, among others) were not analyzed. These circumstances are interesting, owing to their possible interaction with the variables studied. Future studies should therefore collect data on the history of previous orthodontic treatment, in order to better explain variations in the DAI.”, by the new one: “Existing relationships with other caries risk factors described in the literature (such as income, use of fluorides, dietary habits, anxiety about dental treatment, among others) could also be included in future research. Similarly, data on the history of previous orthodontic treatment should be collected by future studies in order to better explain variations in the DAI.” (lines 335-337)
Minor Issues
Comments on Title, Abstract, and Keywords:
- “The title is suitable and accurately explains what the study was about.”
Answer. Thank you very much.
- “The abstract summarizes well the manuscript except for the conclusion that is missing. Please consider adding a conclusion statement, a sentence that indicates your contribution to the science, and what would be the next steps in the oral health area.”
Answer. Now, we have included in the abstract the full conclusion of the manuscript “whereas prevalence of malocclusion and dental plaque was higher among the 12-year-olds, experience of caries was higher among the 15-year-olds. No noteworthy associations were, however, found with socio-demographic variables. Caries and dental plaque were observed to be risk factors for malocclusion in both age groups. This finding has definite clinical implications for prevention of dental caries and oral health promotion in young adolescents.”
- “The information included in the abstract is precise and follows the data of the manuscript. “
Answer. Thanks for your comment.
- “Keywords are suitable. Nevertheless, please capitalize the words that require it.”
Answer. Following the reviewers´ comments, we have carefully reviewed the keywords. We have modified the old keywords “malocclusion; Dental Aesthetic Index; schoolchildren; dental caries; oral hygiene; dental plaque; prevention; oral health; paediatric dentistry” by the new keywords according to both reviewers comments: “dental aesthetic index; schoolchildren; dental caries; dental plaque; prevention; oral health; paediatric dentistry”
Comments on Introduction:
- “The paragraph (lines 47-54) would better fit into the method section as it is a description of the methods. Please consider moving the paragraph.”
Answer. We have carefully reviewed the explanation about DAI, and, as suggested in a previous comment, we have removed some of the DAI information from the introduction, and we have added it on the Data-collection subsection. A brief explanation about Dai was kept in the introduction, in order to correctly introduce and explain the aim of the study. The information was replaced: “The DAI was used to record a number of parameters of occlusal features relating to tooth position, as well as the relationship between maxillary and mandibular arches. The final DAI score classifies malocclusion into 4 categories, each linked to an orthodontic treatment need: a score of ≤25 indicates normal or minor malocclusion (no treatment needed); a score of 26 to 30 represents moderate/definite malocclusion (elective treatment); a score of 31 to 35 indicates severe malocclusion; and a score of ≥36 indicates a very severe (handicapping) malocclusion. Furthermore, any DAI score of ≥31 is considered to require treatment [10].” (lines 120-126)
- “Line 60: Please rewrite the sentence to read: ..., but IT is also important when it comes to assessing treatment priorities.”
Answer. We appreciate your comment. The suggested change has been made on lines 57-58, resulting in “but it is also important when it comes to assessing treatment priorities.”
- “Please rephrase the sentence in lines 69-70 to be more comprehensible.”
Answer. We appreciate the improvement of this sentence, and it has been modified, as you can see in lines 67-68, where the old sentence “Moreover, caries can cause the loss of teeth, and occlusal changes due to this disease have also been related to a higher incidence of malocclusions” has been changed by the new sentence: “Moreover, higher incidence of malocclusion has also been related to missing teeth due to caries.”
- “The aim in the abstract and introduction sections is not congruent. I suggest using the more detailed one in both sections.”
Answer. We have carefully reviewed the aim in both sections, and, as you suggested, the more detailed one was used. We have detailed the aim in the abstract, as you can see in lines 18-20: “The Dental Aesthetic Index (DAI) was determined in 12- and 15-year-old schoolchildren to ascertain the prevalence of malocclusion and to assess its association with dental caries experience, dental plaque accumulation, and socio-demographic variables.”
Comments on Results and Figures:
- “Could you please rephrase the sentence in lines 164-169 to be clearer?”
Answer. Following your indications, we have modified the paragraph for made it clearer. The modification could be seen at lines 169-175, where we have changed the old text “The final sample included 1453 schoolchildren (out of a possible 1843), comprising 868 pupils (418 males and 450 females) aged 12 years and 585 pupils (271 males and 314 females) aged 15 years. Application of the exclusion criteria led to the following losses: absence on the day of the interview (10 and 5 losses in the 12- and 15-year-old groups respectively); and orthodontic treatment at the time (176 and 198 cases in the 12- and 15-year-old groups respectively)”, by the new text: “The final sample included 1453 schoolchildren, comprising 868 pupils (418 males and 450 females) aged 12 years and 585 pupils (271 males and 314 females) aged 15 years. Application of the exclusion criteria resulted in 10 and 5 losses in the groups aged 12 and 15 years, respectively, due to absence on the day of examination; and 176 and 198 losses in the 12- and 15-year-old groups respectively, due to the presence of orthodontic appliances at the time of examination.”
- “Data was presented and reported very nicely and has no further comments.”
Answer. Thank you very much for this nice comment.
Comments on Acknowledgements:
- “The acknowledgments section should be revised. Consider also writing in the past tense.”
Answer. We have revised the acknowledgments section as suggested, and changes were made. We have changed the old paragraph “Our thanks to the Galician Regional Health Authority (Consellería de Sanidade, Dirección Xeral de Saúde Pública). We also appreciate the support staff for their efforts in undertaking the dental examinations and data-collection. We should also like to thank all students and parents for their participation and co-operation with the examination team, and the school staff for their patience and help. Finally, thanks must go to the Department of Psychiatry, Radiology and Public Health, and more specifically to the Preventive Medicine and Public Health Department of the University of Santiago de Compostela” by the new paragraph: “The authors would like to thank the Galician Regional Health Authority (Consellería de Sanidade, Dirección Xeral de Saúde Pública) for the financial support. The assistance provided by the Department of Public Health (University of Santiago de Compostela) and the contribution of all the participants in this study were also greatly appreciated.”
Comments on References:
- “The authors of the manuscript cited relevant references that represent current knowledge in the oral health area. They have also cited a variety of research groups and, therefore, references could be helpful for other researchers. However, references should be cited before full stops or commas. Please revise all the references according to the Journal recommendations. “
Answer. Following your instructions, we have carefully reviewed all the references and we have modified their position in the manuscript. You can see all the changes highlighted throughout the text.

Reviewer 2 Report
Thanks for submitting your manuscript. I enjoyed reading your manuscript but it can be improved by addressing the following points-
- The research should mentioned why it targeted only 12 and 15 years children
- Remove malocclusion from the keywords list
- What do you mean by systematic cross-sectional study? Is there any literature to support this?
- Provide some information on Galician Oral Health survey. Is the data available on public domain?
- How many schools participated in the survey?
- Kindly provide an explanation why the sample size was increased to 10%
- In the discussion, please try to incorporate the following articles
- https://doi.org/10.4103/jispcd.jispcd_333_20
Author Response
Dear Editors and Reviewers,
Thank you very much for your letter and for the reviewers’ comments concerning our manuscript entitled “The Dental Aesthetic Index and its association with dental caries, dental plaque and socio-demographic variables in school-children aged 12 and 15 years”. Those comments are all valuable and very helpful for revising and improving our paper. Based on your comments and requests, we have made careful modifications on the original manuscript.
Reviewer 2
- “The research should mentioned why it targeted only 12 and 15 years children.”
Answer. As described in World Health Organization. Oral Health Surveys. Basic Methods. 4th Edition. Geneva; 1997, 12 and 15-years-old group are considered index ages, and they are recommended for population surveys. In addition, we have analyzed the papers again and have introduced new information that could explain why the study targeted these age groups. You can see this information in lines 69-73: “12- and 15-year-old adolescents represent a very important study group in epidemiological surveys of caries for several reasons, including: easy access to this population at school; the final stage of eruption of permanent teeth; the possibility of analyzing the first years of permanent dentition in the oral cavity; and the beginning of self-made decisions about diet and hygiene [2,9].”
- “Remove malocclusion from the keywords list”
Answer. Following your suggestion, this word has been removed from the keyword list, as you can see in page 1, line 34-35. As suggested in a previous comment, we have also capitalized the old keywords “malocclusion; Dental Aesthetic Index; schoolchildren; dental caries; oral hygiene; dental plaque; prevention; oral health; paediatric dentistry” by the new keywords: “dental aesthetic index; schoolchildren; dental caries; dental plaque; prevention; oral health; paediatric dentistry”
- “What do you mean by systematic cross-sectional study? Is there any literature to support this?”
Answer. We used the word systematic because we carried out several oral health surveys in Galicia (1995, 2000, 2005, 2010) following the same methodology but we have not reviewed the literature systematically. This is the reason why, following your indications, we have changed the old sentence” We conducted a systematic, descriptive, cross-sectional study” by the new sentence: “We conducted a descriptive, cross-sectional study”. You can see these changes in the Materials and Methods section, line 82.
- “Provide some information on Galician Oral Health survey. Is the data available on public domain?”
Answer. The final report of the Galician Oral Health survey has not been published, but several papers have been published with the obtained results:
1.Prevalence and caries-related risk factors in schoolchildren of 12- and 15-year-old: a cross-sectional study.
Obregón-Rodríguez N, Fernández-Riveiro P, Piñeiro-Lamas M, Smyth-Chamosa E, Montes-Martínez A, Suárez-Cunqueiro MM.
BMC Oral Health. 2019 Jun 18;19(1):120. doi: 10.1186/s12903-019-0806-5.
PMID: 31215489 Free PMC article.
- [Oral health in Galician schoolchildren. 1995].
Lorenzo García V, Smyth Chamosa E, Hervada Vidal X, Fernández Casal R, Alonso Meijide JM, Amigo Quintana M, González-Zaera Barreal J, Montes Martínez A, Taracido Trunk M, Cerdá Mota T.
Rev Esp Salud Publica. 1998 Nov-Dec;72(6):539-46.
PMID: 10050604 Spanish.
3.[Dental care in school children in the area of Pontevedra].
Gestal Otero JJ, Smyth Chamosa E, Taracido Trunk M, Cruz del Río JM.
Rev Sanid Hig Publica (Madr). 1987 May-Jun;61(5-6):521-30.
PMID: 3438692 Spanish. No abstract available.
4.[Prevalence of caries in the six-year molar in students at Rio de Pontevedra].
Smyth Chamosa E, Gestal Otero JJ, Taracido Trunk M.
Rev Esp Estomatol. 1987 Mar-Apr;35(2):85-90.
PMID: 3483479 Spanish. No abstract available.
- “How many schools participated in the survey?”
Answer. From the total of secondary schools targeted in Galicia, 485, 60 of them were randomly selected to obtain the sample which we worked with in this study. In each school of these 60 schools, a students´ classroom aged 12 and another students´ classroom aged 15 were selected. You can see this information in the Materials and Methods section, page 3, lines 82-83: “attending a total of 485 schools which imparted compulsory secondary education in north-west Spain.” And line 88-91: “In the sample, one classroom was randomly selected at each school; since the sample included 60 schools, this yielded 60 classrooms of pupils aged 12 years and 60 classrooms of pupils aged 15 years”.
- “Kindly provide an explanation why the sample size was increased to 10%.”
Answer. According to the expert in statistics, the theoretical sample should be increased 10% in order to compensate for the expected missing caused by different reasons. Lines 98-99, we have modified the old text to clarify it, changing the old sentence “The sample size was increased by 10% to offset any expected missing pupils, and to correct for a design effect which we assumed to be 1.5.” by the new one: “The calculated sample size included an additional 10% of individuals to offset any expected missing pupils, and to correct for a design effect which we assumed to be 1.5.”
- “In the discussion, please try to incorporate the following articles https://doi.org/10.4103/jispcd.jispcd_333_20 “
Answer. Following your suggestion, we have tried to incorporate in the discussion section the article titled “The Perception of Undergraduate Dental Students Towards a Clinical Learning Environment at School of Dentistry and Oral Health, Fiji National University, Fiji, however, we do not find the right place to introduce this reference in our manuscript due to the topic is different.
Sincerely yours,

Round 2
Reviewer 1 Report
Major issues:
Comments on Materials and Methods:
- When reporting numbers lower than 10, they should be written with words. Please revise all numbers accordingly.
- All other issues were revised successfully and have no further comments.
Comments on Discussion:
- Regarding the strengths of the study, I suggest corroborating them a bit more as this part provides information about the article’s best quality to the reader.
- I appreciate you revised the sentence in lines 332-333. However, I would suggest inserting one more sentence describing the reason why, i.e., “Firstly, the cross-sectional study design cannot determine whether the caries is the cause or the consequence of malocclusion BECAUSE ...”
- The same applies to lines 335-337. Could you please add an explanation to the text why have not you tried to include these factors?
“Existing relationships with other caries risk factors described in the literature (such as income, use of fluorides, dietary habits, anxiety about dental treatment, among others) could also be included in future research.”
- All other issues were revised appropriately.
Minor issues:
Comments on Title, Abstract, and Keywords:
- The authors of the manuscript have successfully revised all issues and have no further comments.
Comments on Introduction:
- All comments were adequately revised.
Comments on Results and Figures:
- The first sentence indicated to be revised needs revision of all numbers lower than 10 with words.
All other comments, including comments on acknowledgments and references, have been revised accordingly.
Author Response
Dear Editors and Reviewers,
Thank you very much for your letter and for the reviewer's comment on our manuscript entitled "The dental aesthetics index and its association with dental caries, dental plaque and sociodemographic variables in 12- and 15-year-old schoolchildren." Based on your minor comments, we have made careful modifications to the manuscript marked in green. In addition, a native English speaker reviewed the manuscript. Minor changes to English corrections are highlighted in red.
Reviewer 1
Minor issues
- Comments on Materials and Methods.
Reviewer. When reporting numbers lower than 10, they should be written with words. Please revise all numbers accordingly.
Response. Now, we have corrected this mistake (see lines 103 and 120).
Reviewer. All other issues were revised successfully and have no further comments.
Response. Thank you very much.
- Comments on Discussion:
Reviewer. Regarding the strengths of the study, I suggest corroborating them a bit more as this part provides information about the article’s best quality to the reader.
Response. Now, we have included more information regarding the strengths of our study “Another strength in our study was the use of data obtained by trained clinical examiners, which represents greater objectivity than data obtained exclusively through self-completed questionnaires. Likewise, collecting data over several years strengthens our results.” (see lines 330-333).
Reviewer. I appreciate you revised the sentence in lines 332-333. However, I would suggest inserting one more sentence describing the reason why, i.e., “Firstly, the cross-sectional study design cannot determine whether the caries is the cause or the consequence of malocclusion BECAUSE ...” The same applies to lines 335-337. Could you please add an explanation to the text why have not you tried to include these factors?
“Existing relationships with other caries risk factors described in the literature (such as income, use of fluorides, dietary habits, anxiety about dental treatment, among others) could also be included in future research.”
Response. Now, the paragraph of limitations has been changed (green and red color) “Our study is not free of limitations. Firstly, the cross-sectional study design cannot determine whether caries is the cause or consequence of malocclusion because both variables have been analyzed at the same time. Existing relationships with other caries risk factors described in the literature that we have not included in our present research, such as income, use of fluorides, dietary habits, or anxiety about dental treatment, could also be included in future research. Similarly, data on the history of previous orthodontic treatment should be collected by future studies in order to better explain variations in the DAI.” (see lines 334-338)
Reviewer. All other issues were revised appropriately.
Response. Thank you very much.
- Comments on Title, Abstract, and Keywords:
Reviewer. The authors of the manuscript have successfully revised all issues and have no further comments.
Response. Thank you very much.
- Comments on Introduction:
Reviewer. All comments were adequately revised.
Response. Thank you very much.
- Comments on Results and Figures:
Reviewer. The first sentence indicated to be revised needs revision of all numbers lower than 10 with words.
Response. Now, we have corrected this mistake (see line 170).
- Reviewer. All other comments, including comments on acknowledgments and references, have been revised accordingly.
Response. Thank you very much.
Sincerely yours,
